# How to Identify Invasive Candidemia in ICU—A Narrative Review

**DOI:** 10.3390/antibiotics11121804

**Published:** 2022-12-12

**Authors:** Joana Alves, Carles Alonso-Tarrés, Jordi Rello

**Affiliations:** 1Infectious Diseases Department, Hospital de Braga, R. Comunidades Lusíadas 133, 4710-357 Braga, Portugal; 2Microbiology Department Laboratory, Fundació Puigvert, C. de Cartagena, 340, 350, 08025 Barcelona, Spain; 3Departament de Genètica i Microbiologia, Universitat Autònoma de Barcelona, Cerdanyola del Vallès, 08193 Barcelona, Spain; 4Clinical Research in Pneumonia & Sepsis (CRIPS), Vall d’Hebron Institute of Research (VHIR), Pg. de la Vall d’Hebron, 129, 08035 Barcelona, Spain; 5Clinical Research, CHU Nîmes, Rue du Professeur Robert Debré 4, 30900 Nîmes, France

**Keywords:** critical care patients, intensive care units, invasive fungal infection, candidemia

## Abstract

The incidence of invasive fungal infection in ICUs has increased over time, and *Candida* spp. is the most common cause. Critical care patients are a particular set of patients with a higher risk of invasive fungal infections; this population is characterized by extensive use of medical devices such as central venous lines, arterial lines, bladder catheters, hemodialysis and mechanical intubation. Blood cultures are the gold standard diagnosis; still, they are not an early diagnostic technique. Mannan, anti-mannan antibody, 1,3-β-D-glucan, *Candida albicans* germ tube antibody, Vitek 2, PNA-FISH, MALDI-TOF, PCR and T2Candida panel are diagnostic promising microbiological assays. Scoring systems are tools to distinguish patients with low and high risk of infection. They can be combined with diagnostic tests to select patients for pre-emptive treatment or antifungal discontinuation. Candidemia is the focus of this narrative review, an approach to contributing factors and diagnosis, with an emphasis on critical care patients.

## 1. Introduction

Patients admitted to intensive care units (ICUs) have the highest risk of healthcare-associated infection, 19.2% compared to 5.2% on the 2018 European point-prevalence survey [1]; The incidence of invasive fungal infection in ICUs has increased over time, and *Candida* is the most common cause [2,3]. The most frequent invasive fungal diseases in ICU are invasive candidiasis and invasive aspergillosis (among other molds). Invasive candidiasis is mostly manifested as candidaemia [4]. *Candida* spp. is a leading cause of bloodstream infections (BSIs) [5,6,7], and mortality associated with invasive *Candida* infections remains high. Crude mortality can reach up to 50% [5,6,8,9]. Candidemia prolongs hospital stays and increases the costs associated with patient management [6,9].

Among human pathogenic *Candida* spp., *Candida albicans*, *Candida glabrata*, *Candida parapsilosis*, *Candida tropicalis* and *Candida krusei* account for the majority of infections [10]. *C. albicans* remains the most common species causing candidemia, yet non-albicans Candida has been rising [10]. This epidemiological change may be partially explained by the use of antifungals. Other risk factors for infection include previous *Candida* colonization, exposure to broad-spectrum antibiotics, malignancy, surgery or use of intravascular catheters, among others [10,11].

Despite improvements in diagnosis, it remains a challenge in intensive care units. Early diagnosis, source control and timely antifungal therapy are the cornerstone. Scoring systems are tools to distinguish patients at low and high risk of infection in an ICU setting. Scoring systems can be combined with diagnostic tests for optimal utilization. Prompt diagnoses can be made with non-culture diagnostic tools, yet they do not substitute blood cultures. The gold standard for candidaemia diagnosis is *Candida* identification in blood cultures. There are several diagnostic methods used for the rapid identification of *Candida* spp. based on biochemical characteristics or molecular amplification, each one with limitations.

A worrying emergence of resistance in *Candida* spp. in critically ill patients threatens appropriate antifungal therapy [12]. Antifungal stewardship (AFS) is a component of antimicrobial stewardship and has received increasing relevance to optimize the use of antifungal therapy.

Candidemia is the focus of this narrative review, an approach to contributing factors and diagnosis, with an emphasis on critical care patients.

## 2. Candidemia Risk Factors

Advances in medicine allowed us to decrease mortality and prolong life; still, the growing number of immunocompromised patients and associated risk factors explain the increased frequency of candidemia (Table 1).

*Candida* spp. are yeasts, and there are more than 150 species, but only a few causes disease in humans. *Candida* spp. are normal human commensals and can be isolated on the skin, gastrointestinal tract, expectorated sputum or respiratory specimens in intubated patients, female genital tract and in the urine of patients with indwelling catheters. *Candida* spp. becomes pathogenic after immune system defects, either iatrogenic or idiopathic.

Critical care patients are a particular set of patients with a higher risk of invasive fungal infections; this population is characterized by extensive use of medical devices such as central venous lines, arterial lines, bladder catheters, hemodialysis and mechanical intubation [10,11]. All these devices are gateways for colonization and further infection.

### 2.1. Colonization and Infection

Prior *Candida* spp. colonization is an independent risk factor, particularly in patients with multifocal fungal colonization [13]. According to León et al. [13], mortality rate was higher in patients with multifocal colonization, with 50.9% against 26.5% mortality rate in patients with unifocal colonization. Multifocal colonization was defined when *Candida* spp. were simultaneously isolated from various non-contiguous foci [13]. A previous study did not associate colonization with infection risk [14], yet only rectal and/or urine isolates were collected. Pittet et al. [15] demonstrated a higher risk of fungal infection depending on the intensity of colonization—patients colonized at more than two sites.

Etiology has seen major shifts through time towards non-albicans *Candida*, and different *Candida* spp. identification are related to different contributing factors. Candidemia by *C. glabrata* is described in patients with solid organ transplants and with previous antifungal therapy [10,16,17]. *C. parapsilosis* is identified in patients with recent surgery, patients with intravascular devices or parental nutrition [10,18]. *C. parapsilosis* has a particular affinity to intravascular devices due to their adherence ability and biofilm formation [19].

*C. tropicalis* and *C. krusei* were mainly isolated in patients with hematologic malignancies [10,16,17]. Patients on dialysis or with HIV infection were prone to *Candida dubliniensis* infection [10]. *Candida guilliermondii* cases had prior antifungal exposure [10]. *Candida lusitaniae* had solid tumor history and recent surgery [10]. *Candida auris* is an emerging multidrug-resistant yeast in patients with surgery or intravascular devices and with previous antifungal therapy [20] and is prone to cause nosocomial outbreaks [21]. *C. albicans* infection has a lower mortality risk when compared to non-*albicans Candida* [12,22].

The emergence of resistance in *Candida* spp. has raised concern in critically ill patients and threatens appropriate antifungal therapy. *C. glabrata* is the most commonly resistant identified species [12]. The reduced susceptibility to azoles has modified antifungal prescription practices for echinocandins; as a result, selection pressure increased resistance to echinocandins [23,24,25].

### 2.2. Malignancy 

The base of the immune response to invasive candidiasis is a neutrophil function, and neutropenia is a well-recognized risk factor for invasive fungal infection. Monocytes/macrophages also play an important role in protection. Cancer patients have immune defects and a disruption in intestinal mucosal integrity that allows local *Candida* spp. overgrowth and access to the bloodstream. Patients with candidemia have a 30 to 50% cancer prevalence [10,26]. There are a number of cancer diagnoses related to candidemia, such as acute leukemia, lymphoma or myelodysplastic syndrome, allogeneic hematopoietic cell transplantation and graft versus host disease [26,27,28].

Among solid tumors, the risk correlates with *Candida* colonization sites. Patients with gastrointestinal cancer have the majority of cases of invasive fungal infections, followed by genito-urinary cancer [26,27].

### 2.3. Surgery

Bloodstream infections caused by *Candida* spp. are mainly caused through the intestinal barrier, particularly relevant in patients after surgery due to mucosal damage.

The incidence of candidaemia is higher on surgical ICUs when compared to medical ICUs [14,22]. Surgery is a major risk factor and is well-proven when involving the gastrointestinal tract [14,26]. This association is explained by *Candida* spp. colonization in this site. Patients submitted to upper gastrointestinal tract surgery or the presence of gastroesophageal junction leakage is a risk factor for *Candida* infections [29]. Patients under thoracic surgery also had a risk for candidaemia, while trauma and neurosurgical cases had a lower risk [14]. 

Surgical patients usually recover with hyperalimentation fluids, a relevant risk for candidemia. Total parental nutrition has been associated with a higher risk for candidaemia than peripheral parental nutrition (OR 26.8 vs. 20.0) [30].

### 2.4. Catheter-Related Bloodstream Infection

Catheter-related Bloodstream Infection (CRBSI) is defined as occurring 48 h before or after catheter removal and positive culture with the same microorganism of either quantitative CVC culture ≥ 10^3^ CFU/mL or semi-quantitative CVC culture > 15 CFU or BSI occurring with or without catheter removal, and quantitative blood culture ratio CVC blood sample/peripheral blood sample > 5 or differential delay of positivity of blood cultures (CVC blood sample culture positive two hours or more before peripheral blood culture) or positive culture with the same microorganism from pus from insertion site [31].

Skin colonization is the first step for invasive candidiasis. Devices disrupt the physical barrier of the skin and mucous membranes allowing the fungus to access the blood. Cardiovascular invasive procedures and the presence of intravascular catheters are common risk factors [10,32].

Patients admitted to intensive care units have the highest risk of healthcare-associated infection (HAI), 19.2% compared to 5.2% on the 2018 European point-prevalence survey (PPS) [1]. Bloodstream infections were the fourth most frequently reported HAI, 10.8% in PPS [1]. *Candida* spp. was one of the 10 most frequently isolated microorganisms [7]. In ICU, BSIs are the third most common site of infection and the highest infection-associated mortality [3]. Among patients with positive microbiological cultures, 16% had a fungal microorganism [3].

Nosocomial BSIs often are related to the presence of a catheter; therefore, ICUs have higher rates of catheter-related BSI.

### 2.5. Sepsis

Septic shock in the setting of candidemia was believed to occur less than bacteremia septic shock, but in a 2016 published EUROBACT study [32], 39.6% of patients admitted with fungemia presented with septic shock against 21.6% of bacteremia patients. Candida septic shock carries a high crude mortality, reported being in the range of 36 to 61% [3,33,34,35]. Risk factors associated with higher mortality in ICU are failure of source control, delay in antifungal therapy, and increasing APACHE score [35]. Patients with vasopressors who underwent renal replacement therapy or positive ventilatory support present an increase in the volume of distribution and are exposed to antifungal underdosing, and therapeutic drug measurement (TDM) is advised.

### 2.6. Broad-Spectrum Antibiotics

Antibiotics are responsible for changes in endogenous microbial flora, which allows fungal overgrowth on site. The use of broad-spectrum antibiotics is believed to be a risk factor for *Candida* spp. infection [36], particularly when more than two drugs are used [11]. In a recent retrospective study on catheter-related *C. parapsilosis* BSI, patients with prior use of more than three antibiotics had seven times greater risk of candidemia [37]. Quinolones and third generation cephalosporins were the mainly used antibiotics associated with Candida BSI in intensive care units [38]. 

Antibiotic consumption has decreased between 2019 and 2020 (consequences of the SARS-CoV-2 pandemic), yet antibiotic consumption is higher in critical care compared to infirmary patients. On the other hand, antifungal consumption has increased, possibly due to an increase in fungal co-infection in patients with COVID-19 and corticosteroid use [39]. Previous use of antimicrobial therapy must be evaluated in all patients with suspected *Candida* spp. infection. 

## 3. Diagnostic Approach

Invasive candidiasis (IC) includes candidaemia and deep-seated tissue candidiasis. Candidaemia is the most common identity in ICU versus deep-seated candidiasis [4]. Diagnosis of candidaemia is challenging due to its similarity with other infections. The selection of patients with *Candida* isolates (colonization) who have the risk of invasive infection is a decisive step. Specimens often can be colonized with *Candida* species in respiratory samples (sometimes representing oral flora contamination), urine (genital contamination), surgical drainages (external skin contamination) or indwelling catheters, and they must not be interpreted as true infection. Diagnostic misinterpretations can lead to unnecessary antifungal use, antifungal resistance, ignoring the true pathogenic organisms and increased healthcare costs. Candidaemia is defined by the presence of *Candida* species in the blood. Diagnosis is established by blood culture, ideally three different sets of two bottles with 10 mL in each bottle (total of 60 mL) [40]. 

Signs of invasive candidiasis might be scarce, and the diagnosis is usually late in the course of the ICU stay. The median time between the onset of infection and antifungal therapy can be up to eight days [34]. Early diagnosis is challenging, and a high index of suspicion is the baseline approach to the patient with risk factors for invasive fungal infection. 

A probable diagnosis of IC requires the presence of a risk factor, clinical criteria and mycological evidence [41]. Mycological criteria are based on cytology, direct microscopy or culture in a sterile site, or detection of B-1,3-D-glucan (BDG) detected in at least two consecutive serum samples or a *Candida* spp. identification with the T2Candida panel [41,42]. Possible invasive candidiasis is defined as a patient with a risk factor and clinical syndrome without mycological criteria [41].

Bassetti et al. [4] proposed a definition of probable IC for critically ill patients where some host factors were adjusted to ICU: impaired gut wall integrity (patients with recent abdominal surgery, recurrent intestinal perforations), impaired cutaneous barriers to bloodstream infection (presence of central vascular access device, ECMO cannula or hemodialysis) and Candida colonization (recovery of *Candida* spp. in cultures from two or more sites).

### 3.1. Culture

The standard of care for definite diagnosis is the isolation of *Candida* in blood cultures (BC), sill it is not an early diagnostic technique. Sensitivity of BC to detect *Candida* ranges from 50 to 71%. Still, it can be lower in neutropenic patients [43,44]. *Candida* isolation can take between two and three (some until eight) days to grow [45]. Time to positivity is different between species. *C. glabrata* grows slower than *C. albicans* [46]. Blood cultures can be negative in patients with antifungal drug exposure, and sensitivity can be increased when the volume of the complete set of blood cultures is 60 mL. In patients with probable invasive candidiasis, the recommended frequency of blood culture collection is daily [44]. Non-culture diagnostic tools do not substitute blood cultures. They can only be combined for earlier intervention. After organism identification in BC, an antifungal susceptibility test is required to guide the management of candidaemia and oral azole de-escalation due to the emergence of resistance to azoles and echinocandins, significant in *C. glabrata*.

### 3.2. Serum Biomarkers

Biomarkers are essential tools for early diagnosis; still, despite extensive research, they are not validated to distinguish colonized patients from patients with fungal infections. Available biomarkers, such as mannan, anti-mannan antibody and 1,3-β-D-glucan (BDG), have been developed to improve and anticipate the detection of invasive disease prior to microbiological confirmation.

Mannans are a main cell wall component of *Candida* spp. and are used to detect *Candida* infections [47]. The combined detection of mannan and anti-mannan antibody increases the sensitivity from 58% to 83% and specificity from 59% to 86% [48,49]. In the ICU setting, they have a high negative predictive value, which is particularly useful in excluding invasive *Candida* infections, especially after five days of unnecessary antifungal therapy [33]. These biomarkers can predict infection prior to blood cultures. They are an important tool in reducing the diagnosis time yield or reducing the use of antifungal agents. For non-*albicans* bloodstream infections, such as *C. parapsilosis* and *C. krusei*, antigen and antibody detection have lower sensitivity [50].

BDG is a pan-fungal diagnostic test. BDG is a cell wall component of *Candida* and other fungi (such as *Aspergillus*, *Pneumocystis jiroveci* and others) with a high diagnostic sensitivity of 75–80% and specificity of around 80% [49] Odabasi et al. [51] reported positive BDG result up to 10 days before clinical diagnosis in patients with proven or probable invasive fungal infection.

To optimize BDG performance, two consecutive positive results are required. A meta-analysis to evaluate the accuracy of BDG on ICU patients by Haydour et al. [52] reported 80% of sensitivity but low specificity (only 60%). Patients with albumin, renal replacement therapies with cellulose membranes, intravenous immunoglobulin and concomitant BSI, may have false positive BDG [43]. BDG can be used to withdraw unneeded antifungals due to high negative predictive value (NPV) in ICU [53,54].

Antibodies against *C. albicans* germ tubes (CAGTA) is an immunofluorescence assay that detects responses against a hyphal protein expressed during tissue invasion and biofilm formation [49]. Sensitivity and specificity are variable and higher in candidemia [55,56,57]. Still, false positives can occur in the presence of concomitant bacterial infection [58]. Serology tests usually take time to be positive, normally more than 15 days after the first encounter of the microorganism antigen with the host. Some researchers have started to study the kinetics of these antibodies [59]. Nevertheless, more knowledge about the time to positivity after the initiation of candidemia of this promising test is needed to decide if it is useful at the beginning of the clinical episode or only after some days. It might be more useful to rule out candidemia in already treated patients and withdraw unnecessary antifungal empirical treatment [33].

A positive biomarker report does not provide a diagnosis yet raises the probability of infection. Simultaneous use of different biomarkers can improve the negative predictive value or avoid false positive results. The combination of CAGTA and BDG improved the NPV to 95 to 97% for proven invasive candidiasis in some ICUs [48,60].

Biomarkers can be expensive and time-consuming. They should probably not be used widely for the potential risk of extra antifungal consumption. Additionally, the use of a biomarker-based strategy in the ICU demonstrated increased early discontinuation of empirical antifungal therapy without a negative impact on outcomes [48,61].

### 3.3. Molecular Biology

Polymerase Chain Reaction (PCR) performed in blood samples have the highest sensitivity, 90–95%, and specificity, 90–92%. PCR shortens the time to a diagnosis, yet the interpretation is heterogenic, and colonized patients may have a positive PCR [55,62]. The need to follow a strict aseptic technique to obtain the blood for this test is as important, if not more, than for the routine blood cultures because PCR detects very small quantities of the genetic material of either viable or non-viable microorganisms.

The detection limit of PCR is under 10 CFU/mL; still, if the number of *Candida* CFU/mL in the blood is under the threshold, like in an early set of the disease, the test might be negative. Pfeiffer et al. [45] reported CFU/mL of ≤1 on half of *Candida* spp. blood cultures, particularly in patients with candidemia by *C. glabrata*. Low organism burden was associated with neutropenic patients, recent major surgery, end-stage live disease, renal replacement therapy, interrupted gastrointestinal tract and candidemia from the abdominal site [45]. The five most common pathogenic *Candida* spp., such as *C. albicans*, *C. glabrata*, *C. parapsilosis*, *C. tropicalis* and *C. krusei*, can be targeted by commercial multiplex PCR kits. Specificity is superior in molecular amplification techniques over BDG and CAGTA, and, as a consequence, better positive predictive values can be achieved [55].

The BioFireFilmArray BCID assay identifies 24 organisms (19 bacteria and 5 most common *Candida* species) by multiplex PCR from positive blood cultures [63], with a sensitivity of 100% and results in one hour [64]. An updated version of the panel—BCID2 identifies 33 species, including 7 fungi, 6 *Candida* spp. (*C. albicans*, *C. glabrata*, *C. parapsilosis*, *C. tropicalis*, *C. krusei* and *C. auris*) [65]. 

A novel nanodiagnostic test, T2Candida Panel, is a PCR-based assay that detects *Candida* within whole blood through mechanical lyses of cells and DNA amplification later detected by amplicon-induced agglomeration of super magnetic particles and T2 magnetic resonance measurement. Sensitivity and specificity are 89–91% and 98–99% [66,67]. The limit of detection is 1 CFU/mL [67], and the limit of blood cultures is 1 CFU/60 mL of blood, usually obtained in a routine set of three 20 mL samples. An important feature is the availability of the result in 3–4 h directedly from whole blood. The T2Candida Panel reports the five most common *Candida* spp. as a positive or negative result. Results are reported based on susceptibility to fluconazole and divided as *C. albicans*/*C. tropicalis*, *C. parapsilosis*, and *C. krusei*/*C. glabrata*. An important benefit of this novel test is the higher sensitivity on follow-up analysis versus blood cultures, demonstrated in already Candida BSI patients either with neutropenia or in patients receiving prior antifungal therapy [68], probably because it is a genetic amplification assay that detects non-viable yeasts. Positive blood cultures and positive T2Candida in candidemia follow-up samples had higher mortality (42% vs. 5% when they were negative) [68].

All these molecular tests do not discriminate colonization or past infection from ongoing true infection, and diagnostic stewardship is advised to avoid over-diagnosis of true active fungal infections.

### 3.4. Other Methods

Conventional methods of identification are based on morphological and biochemical *Candida* characteristics. They have low identification accuracy and are time-consuming [69]. Biochemical methods based on automated systems improve etiology diagnosis. VITEK 2 is an automated identification system capable of antifungal susceptibility testing after positive blood cultures. VITEK 2 has sensitivity and specificity above 95% for common *Candida* spp. [70]. Misidentification of uncommon *Candida* spp. [69,71,72] were reported, particularly *C. guilliermondii*. Peptide Nucleic Acid in Situ Hybridisation Yeast Traffic Light system (PNA-FISH YTL) and matrix-assisted laser desorption/ionization time-of-flight mass spectrometry (MALDI-TOF MS) rapidly identifies *Candida* spp., still only after positive blood cultures.

Although high sensitivity (96 and 99%), PNA-FISH YTL cannot distinguish between *C. albicans* and *C. parapsilosis* or between *C. glabrata* and *C. krusei* [73,74]. MALDI-TOF MS has a 56 to 73% sensitivity and differentiates distinct and related species [75,76]. *C. auris*, often misdiagnosed, can be identified on MALDI-TOF MS [77]. MALDI-TOF MS requires pure growth of the organism on artificial media. After the isolation of the organism, MALDI-TOF takes 10 to 15 min to identify. MALDI-TOF MS can be used for antifungal susceptibility tests. This assay reports fluconazole/caspofungin/anidulafungin resistance for *C. albicans* and *C. glabrata* [76].

The Accelerate PhenoTest^TM^ BC Kit detects *C. albicans* and *C. glabrata* (in addition to identification and rapid antimicrobial susceptibility testing of gram-positive and gram-negative bacteria in positive blood cultures). This assay had 100% and 99% sensitivity and specificity for *C. albicans*, while *C. glabrata* had 80% and 97%, respectively [78].

## 4. Scoring Systems

Since the delay in antifungal therapy is associated with increased mortality in patients with candidemia [79], the selection of the right patient at risk of invasive fungal infection is essential. Antifungal therapy should be considered in intra-abdominal postoperative ICU patients with risk factors for invasive *Candida* infection who present fever of unknown cause (or persistent clinical signs of sepsis) and positive *Candida* serum biomarkers [80]. Early pre-emptive or prophylaxis therapy has been suggested based on scoring systems to select high-risk patients and through the existing diagnostic techniques (Figure 1). Clinical scores are essential tools to distinguish patients at low and high risk of infection and may reduce costs and unnecessary use of antifungals. 

Since colonization status is an independent risk factor and multifocal colonization predictor of a higher risk of invasive *Candida* infection, a colonization index was suggested first by Pittet et al. [15] and then evaluated in a prospective cohort [35] with surgical patients admitted to ICU. Patients with a corrected colonization index (CCI) above 0.4 received preemptive antifungal therapy, and a significant decrease in the acquisition of proven candidiasis was demonstrated. No patients with a CCI under 0.4 developed proven candidiasis.

Ostrosky-Zeichner et al. [81] described three predictive rules (one major criterium and two minor criteria) for candidaemia in an intensive care setting with any antibiotic use, presence of CVC, any surgery, immunosuppressive therapy, pancreatitis, total parental nutrition, dialysis and steroid use. Patients selected by rule 3 had a higher rate of infection, yet only one-third of patients with candidaemia were identified. Another study reported higher sensitivity [82]. 

Candida Score [13] is a simple bedside scoring system. It was developed based on previous colonization status (multifocal) and clinical predisposition: surgery, total parental nutrition and severe sepsis. This score is validated for critically ill patients admitted to an ICU and had 81% sensitivity and 74% specificity. A score above 2.5 in an intra-abdominal postoperative ICU patient was suggested for consideration of starting pre-emptive antifungal agents, and under 2.5 strongly decreases the chance of infection. 

Another scoring system was evaluated in intra-abdominal surgical patients: Dupont et al. [83] demonstrated an 84% sensitivity and 50% specificity with a score based on the female gender, the upper gastrointestinal origin of peritonitis, cardiovascular failure and use of antibiotics. Basseti et al. [84] proposed an algorithm considering abdominal surgical patients versus non-abdominal surgery/medical, colonization status and Candida score and biomarkers to address the need for antifungals; the efficacy and mortality should be validated. 

These scores are targeted to abdominal surgical patients admitted to ICU; these scores are not validated for medical patients admitted to ICU or patients with malignancy. The optimal utilization of risk factors or serum biomarkers or score systems is yet to be known. Its efficacy and influence on mortality call for randomized controlled trials. 

## 5. Antifungal Stewardship

Accompanying the rise of invasive fungal infections, there is a worldwide rise of resistance of *Candida* to azoles and echinocandins [24]. The resistance to azoles has increased the use of echinocandins to treat *Candida* infections; as a consequence, exposure to echinocandins has led to reduced susceptibility [24]. The emergence of fungal resistance has an impact on patient outcomes. There is a call for active stewardship to ensure responsible use and minimize the development of resistance. 

Antimicrobial stewardship refers to “a coherent set of actions which promote using antimicrobials responsibly” [85], which includes optimizing antibiotics, antivirals and antifungals. Antifungal stewardship improves antifungal use, patient care and outcomes [85,86,87,88,89]; AFS interventions reduce the time to antifungal prescription [87]. AFS should be included as part of the AMS program, depending on the ICU setting. In an ICU where the patient population already has a degree of complexity, when fungal infections are frequent and the use of antifungal agents is increasing, AFS is required.

Diagnostic stewardship is a fundamental step in stewardship programs. There are three important issues in diagnosis stewardship improving time to diagnosis, appropriate antifungal based on timely antifungal susceptibility tests and antifungal discontinuation in selected patients in intensive care units with negative biomarkers or negative PCR.

A recent core intervention for Antifungal Stewardship was published [90] with the recommendation to use both fungal culture and non-culture-based tests. Biomarkers and molecular amplification techniques reduce the time yield for diagnosis with the highest sensitivity [48,61,90]. However, they do not represent confirmation of an active infection, and the interpretation is not equivalent to the antifungal requirement.

Empirical antifungal therapy based only on clinical signs and risk factors leads to the unnecessary use of antifungals [87]. Antifungal discontinuation can be based on the high negative predictive value that can be achieved by using non-culture-based tests but nevertheless still requires an individualized patient approach.

Rautemaa-Richardson et al. [91] evaluated the use of BDG testing on antifungal discontinuation. This strategy reduced echinocandin consumption by 39%. Ito-Takeichi et al. [92] implemented an antifungal stewardship intervention based on BDG and reported a significant reduction of antifungals and better outcomes in patients with proven candidaemia.

Appropriate prescription of antifungal drugs depends on timely *Candida* isolation and susceptibility testing; AFS strategies reduce the time to start antifungal therapy [87]. There are several different diagnostic methods used for the rapid identification of *Candida* spp. MALDI-TOF MS, PNA-FISH and Multiplex PCR provide results within minutes to a few hours, only they are blood-culture-based methods. The T2Candida panel is not a blood-culture-based test, reducing the time yield for diagnosis [67,93,94]. Gill et al. [94] evaluated the discontinuation of antifungals, and only 3% had the development of proven candidemia after discontinuation.

Among different AFS interventions to improve patient outcomes with candidaemia, diagnostic stewardship for prompt diagnosis is fundamental for the appropriate and cost-effective treatment of patients at risk of invasive candidiasis. Antifungal therapy for prophylaxis of cannula or indwelling catheter insertion is not recommended.

## 6. Conclusions

Critical care patients are a particular set of patients with a higher risk of invasive fungal infections. This population is characterized by extensive use of medical devices such as central venous lines, arterial lines, bladder catheters, ECMO and renal replacement therapy cannulas and tracheal intubation. Septic shock in the setting of candidemia is a reality in critically ill patients with high crude mortality. Available biomarkers, such as mannan, anti-mannan antibody, 1,3-β-D-glucan and *Candida albicans* germ tube antibody, have been developed to improve and anticipate the detection of invasive disease or withdraw unnecessary empirical antifungal treatment. The use of a biomarker-based strategy in the ICU demonstrated increased early discontinuation of empirical antifungal therapy without a negative impact on outcomes. Still, biomarkers tests should probably not be used widely for the potential risk of extra antifungal consumption. Recent advances in molecular biology shorten diagnostic time yield with high sensitivity and specificity. There are a number of commercial multiplex PCR tests to target the five most common pathogenic *Candida* species after positive blood cultures and a novel nanodiagnostic panel, the T2Candida panel, that detects *Candida* directly from a whole blood sample. Scoring systems are tools to distinguish patients at low and high risk of infection; these scores are targeted to abdominal surgical patients admitted to ICU. Scoring systems can be combined with diagnostic tests to select patients for pre-emptive treatment. Still, the optimal utilization of risk factors and score tools or serum biomarkers is yet to be known. Its efficacy and influence on mortality call for randomized controlled trials.

Antifungal stewardship is fundamental to optimize antifungal therapy and, consequently, patient care and outcomes improvement; diagnostic stewardship is the core strategy to reduce time yield to diagnosis and timely antifungal susceptibility test. Antifungal discontinuation based on non-culture-based tests is reserved for the right clinical setting, such as the intensive care unit.

In summary, appropriate antifungal therapy is a determinant of survival in critically ill patients with susceptible Candida infections. Safe implementation requires a smart strategy to avoid both delays in starting antifungal therapy and avoid over-prescription (for colonization or contaminated specimens). Renal function, the daily dose administered, and the site of infection are determinants of the right prescription. Optimized drug dosing and diagnosis should be considered core priorities for improving clinical outcomes for critically ill patients with fungal infections.

## Figures and Tables

**Figure 1 antibiotics-11-01804-f001:**
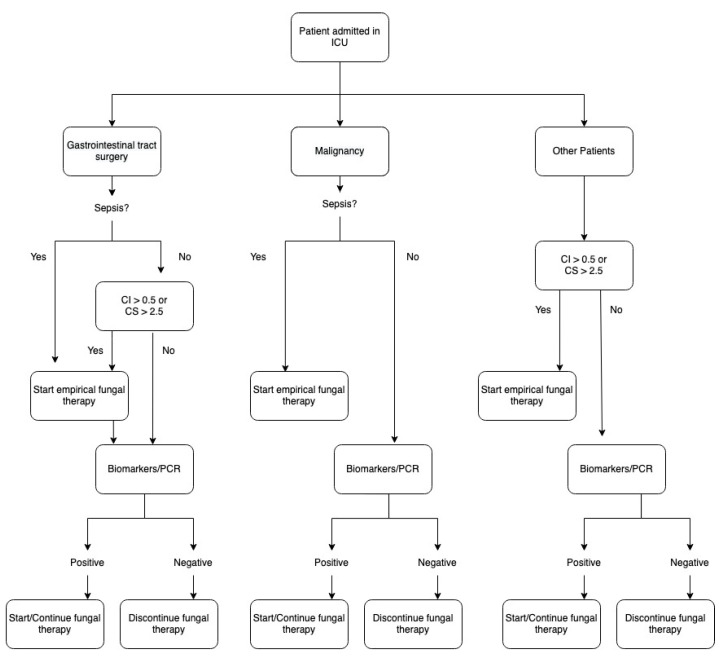
Proposed Algorithm for pre-emptive antifungal therapy on ICU patients who underwent intra-abdominal surgery.

**Table 1 antibiotics-11-01804-t001:** Risk Factors for Invasive Candidemia.

Risk Factors
1. Major Risk Factors
Intravascular devices
Recent surgery (particularly abdominal surgery)
Broad-spectrum antibiotics/antifungals
Immunosuppressive therapy (corticosteroids and chemotherapy)
Malignancies (solid tumors and hematologic)
*Diabetes mellitus*
2. Other Risk Factors
Hyperalimentation fluids
Previous ICU stay
Mechanical ventilation
Urinary catheterization
Prior *Candida* colonization/infection
Concomitant bacterial infections
Solid organ transplant patients
Hemodialysis
HIV-associated low CD^4+^ T cell counts

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
