# Peer review of "How to Identify Invasive Candidemia in ICU—A Narrative Review"

_antibiotics, 2022, doi:10.3390/antibiotics11121804_

Round 1

Reviewer 1 Report

The review summarizes major risk factors for invasive candidaemia and describes current methods for its diagnosis. The paper is well written and fits well for the themed issue in the journal Antibiotics

Suggestions for improvement/clarification:

The term „antimicrobial stewardship“ could be explained a bit when first mentioned

Line 69 “Candida organisms” consider rewording to “Candida strains” or “Candida species”

Line 74 “particularly” should read “particular”

Line 94: “C. parapsilosis has particularly affinity to intravascular devices, parental nutrition and recent surgery” here some rewording/explanation might increase clarity. What is specifically meant with affinity to parental nutrition and recent surgery?

Line 105 “specie” should read “species”

Line 127: “ when involves” should read “when involving the..”

Line 172: “candidaemia is the most common identity in ICU” please clarify, meaning of “identity” unclear in this context

Line 186: “positive T2Candida” further brief explanation required

Line 216 “anti-mannan” should read “anti-mannan antibody”

Line 206: “antifungal susceptibility test” is mentioned in the part describing culture based diagnosis of IC. Since the target journal is “Antibiotics”, it might be appropriate to mention in one or two sentences which antifungal therapy options are commonly employed.

Line 278ff: The mechanism of the mentioned T2 Candida panel test could be described in some more detail. In its current state of description the assay principle doesn’t become entirely clear.

Reviewer 2 Report

     The review written by Alves et al. provides updated information on diagnosing invasive infections with Candida spp. The article has good overall clarity and provides easy-to-understand and useful information on the available diagnosis tool for a condition associated with high mortality rates.

1.      Punctuation and the correct writing of microorganisms names

     The manuscript contains multiple punctuation mistakes (e.g. the over usage of “;” instead of finishing a sentence) and the abbreviations are not properly used. A few examples:

- Candida species should be abbreviated  “Candida spp.” (please be consistent in using abbreviations);

- Line 44: the first time a fungus is mentioned in the text please write the full name (e.g. Candida albicans). Afterward, the abbreviation is acceptable (C. albicans). Please revise the whole manuscript and for all Candida spp. (e.g. Lines 104-105 and many others)

- “Candida” should always be capitalized (see Line 46, Line 155, Line 194, etc.);

-  because of the high number of non-albicans species, “non-albicans” is a group;

2.      The introduction method needs rephasing, as it is the least well-written section of the manuscript

3.       Table 1 – the risk factors are not enough for a table, in this form. Eventually, you can add numbers or organize the risk factors based on their predictive value.

4.      “Catheter-related bloodstream infection” – please add the microbiological diagnosis criteria for  catheter-related bloodstream infection

5.       The section “broad spectrum antibiotics” fails to provide enough information on the subject. It can be extended.

6.       I particularly enjoyed the “Molecular Biology” and “Other methods” sections, as these sections are bringing valuable information in Clinical Microbiology, although there are other diagnostic methods that have been overlooked (other automated systems of identifications, for example)

7.      “Antifungal stewardship” – please include the indications for performing the AFS. The section mostly focuses on the importance of AFS. Please add a paragraph on the laboratory methods that can be used and their limitations.

The paragraph of T2Candida (line 381) is repeating information, in a slightly different form.
